# Stereomeric Lipopeptides from a Single Non-Ribosomal Peptide Synthetase as an Additional Source of Structural and Functional Diversification in *Pseudomonas* Lipopeptide Biosynthesis

**DOI:** 10.3390/ijms241814302

**Published:** 2023-09-19

**Authors:** Penthip Muangkaew, Vic De Roo, Lu Zhou, Léa Girard, Catherine Cesa-Luna, Monica Höfte, René De Mot, Annemieke Madder, Niels Geudens, José C. Martins

**Affiliations:** 1Organic and Biomimetic Chemistry Research Group, Department of Organic and Macromolecular Chemistry, Faculty of Science, Ghent University, B-9000 Ghent, Belgium; penthip.muangkaew@ugent.be (P.M.); vic.deroo@gmail.com (V.D.R.); annemieke.madder@ugent.be (A.M.); 2NMR and Structure Analysis Unit, Department of Organic and Macromolecular Chemistry, Faculty of Science, Ghent University, B-9000 Ghent, Belgium; 3Laboratory of Phytopathology, Department of Plants and Crops, Faculty of Bioscience Engineering, Ghent University, B-9000 Ghent, Belgium; lulzhou.zhou@ugent.be (L.Z.); monica.hofte@ugent.be (M.H.); 4Centre of Microbial and Plant Genetics, Faculty of Bioscience Engineering, Katholieke Universiteit Leuven, B-3001 Heverlee, Belgium; lgv.microbiology.consulting@gmail.com (L.G.); catherine.cessa@gmail.com (C.C.-L.); rene.demot@kuleuven.be (R.D.M.)

**Keywords:** *Pseudomonas entomophila*, cyclic lipodepsipeptide, entolysin, stereochemistry, non-ribosomal peptide synthetase, E/C-type domain

## Abstract

In *Pseudomonas* lipopeptides, the D-configuration of amino acids is generated by dedicated, dual-function epimerization/condensation (E/C) domains. The increasing attention to stereochemistry in lipopeptide structure elucidation efforts has revealed multiple examples where epimerization does not occur, even though an E/C-type domain is present. While the origin of the idle epimerization in those E/C-domains remains elusive, epimerization activity has so far shown a binary profile: it is either ‘on’ (active) or ‘off’ (inactive). Here, we report the unprecedented observation of an E/C-domain that acts ‘on and off’, giving rise to the production of two diastereoisomeric lipopeptides by a single non-ribosomal peptide synthetase system. Using dereplication based on solid-phase peptide synthesis and NMR fingerprinting, we first show that the two cyclic lipopeptides produced by *Pseudomonas entomophila* COR5 correspond to entolysin A and B originally described for *P. entomophila* L48. Next, we prove that both are diastereoisomeric homologues differing only in the configuration of a single amino acid. This configurational variability is maintained in multiple *Pseudomonas* strains and typically occurs in a 3:2 ratio. Bioinformatic analysis reveals a possible correlation with the composition of the flanking sequence of the N-terminal secondary histidine motif characteristic for dual-function E/C-type domains. In permeabilization assays, using propidium iodide entolysin B has a higher antifungal activity compared to entolysin A against *Botrytis cinerea* and *Pyricularia oryzae* spores. The fact that configurational homologues are produced by the same NRPS system in a *Pseudomonas* strain adds a new level of structural and functional diversification to those already known from substrate flexibility during the recruitment of the amino acids and fatty acids and underscores the importance of complete stereochemical elucidation of non-ribosomal lipopeptide structures.

## 1. Introduction

Cyclic lipodepsipeptides (CLiPs) form a structurally diverse group of secondary metabolites that are generally produced by different genera of soil-borne bacteria, such as *Streptomyces*, *Bacillus* and *Pseudomonas* [1,2]. CLiPs display a functional diversity in nature, where they act as important mediators in the chemical ecology of their producers. This includes typical prokaryotic processes, such as bacterial motility, swarming, quorum sensing, biofilm formation, etc. [3,4,5]. Equipped with antimicrobial properties, CLiPs can also be deployed to control competitors in their environment [6,7]. Beyond a fundamental microbiological in/terest in their biosynthesis and role in the chemical ecology of their producers, they feature a broad application potential in various domains. In agriculture, interest has focused on developing these compounds into useful biocontrol and biostimulation agents [8]. The antimicrobial properties, combined with the concerns regarding mounting antibiotic resistance, have spurred research into their use as lead compounds for antibiotics [9,10]. Furthermore, the activity profile of CLiPs is increasingly broadening, as recent findings have reported on their insecticidal [11,12] and anti-carcinogenic activities [13,14].

CLiPs are produced by non-ribosomal peptide synthetases (NRPSs), which are large multienzyme machineries that produce peptides with large structural and functional diversity [15,16,17,18]. These enzymes possess distinct catalytic modules where the growing peptide chain is initiated, then elongated, modified and ultimately released. The number and order of modules in the NRPS is, in most cases, consistent with the amino acid sequence in the peptide moiety, an observation referred to as the co-linearity rule [19]. Typically, a module involved in the elongation of a peptide by one amino acid consists of three separate domains in the following order: the condensation (C) domain, the adenylation (A) domain and the thiolation (T) domain (alternatively referred to as the peptidyl carrier proteins (PCPs)), resulting in a CAT architecture for each module [20]. In short, the A-domains are responsible for the selection of an individual amino acid substrate from the metabolite pool, followed by its activation through adenylation and subsequent transfer to the T-domain of the same module, generating a reactive thioester in the process. Following this, the C-domain of the subsequent module catalyzes amide bond formation between the amino group of its T-tethered amino acid and the T-tethered thioester of the previous module, transferring the growing peptide chain in the process. Subsequent iterations by the following CAT modules result in peptide elongation, before cyclisation by tandem transesterification (TE) domains. The N-terminal acyl chain is introduced by the first module of the NRPS by a C-domain subtype, called C_start_-domain or lipo-initiation domain, that links a fatty acid to the first amino acid. The prevalence of D-configured amino acids in the resulting peptide metabolites results from the action of so-called epimerization or E-domains. These dedicated tailoring domains catalyze the epimerization of L-amino acids introduced by the preceding module.

The NRPS assembly line can demonstrate some flexibility in the substrates recruited by the A-domains, resulting in the production of a handful of structural analogues [17,21]. Consequently, aside from the main or ‘major’ metabolite that is produced by the bacterium, typically some ‘minor’ homologues or congeners are biosynthesized, as well. Typically produced in smaller quantities, they present a single amino acid or fatty acid variation compared to the major CLiP. For example, multiple minor homologues of massetolide A have been described from the same bacterium, whereby amino acid differences occur at position 4 and/or 9 and the identity of the fatty acid [22]. Moreover, in most cases, the alternative amino acid is structurally related to the amino acid that is normally selected by the A-domain, for example, isoleucine versus leucine/valine, or aspartic acid versus glutamic acid [5]. Flexibility of the initial C^start^-domain leads to variability in the acyl chain.

*Pseudomonas* CLiPs feature a majority of D-configured amino acids, yet in stark contrast with NRPS systems of other genera, their NRPSs are devoid of epimerization domains [5]. Instead, the D-configuration of the amino acids is generated by dedicated, dual-function epimerization/condensation (E/C) domains [23]. These E/C-domains not only extend the growing peptide chain by an additional amino acid, but also catalyze the L-to-D conversion of the preceding amino acid in the newly elongated chain. The precise order of dual-function E/C-domains and condensation only, so-called ^L^C_L_-domains (which leave the configuration unchanged), in the NRPS cluster will thus determine the location of D- and L-configurations in the sequence of the produced CLiP. Since ^L^C_L_- and E/C-domains contain signature sequences that can be recognized through bioinformatic tools, such as NaPDoS and antiSMASH [24,25], establishing their presence by analysis of the biosynthetic gene cluster (BGC) is often used to subsequently predict which positions in the final CLiP sequence will feature a D-amino acid [26]. However, as shown by us and others, elevating the prediction to certainty should be avoided, as an increasing number of structure elucidations with explicit determination of configuration indicate that not all identified E/C-domains effectively epimerize the preceding amino acid [27,28,29]. To date, comparative analysis of epi-active and epi-inactive E/C-domains have not revealed unambiguous correlations with specific variations in the amino acid sequence of the active site in dual E/C-domains, thus preventing fail-safe D/L-configurational attribution based on genomic data alone. Therefore, full determination of CLiP structure still necessitates the use of extensive chemical structure analysis and, in some cases, chemical synthesis [30]. This need is now further underscored by the discovery of configurational flexibility within a single NRPS assembly line.

More specifically, as part of our efforts to structurally validate existing or elucidate new *Pseudomonas* CLiPs, we report on our investigation of CLiPs produced by various plant-associated *Pseudomonas entomophilia* strains and their comparison with the entolysins produced by the type strain *P. entomophilia* L48^T^ extracted from the fruit fly [31,32]. Using our NMR fingerprint-based approach, we show that in all cases, two CLiPs are produced, with the major CLiP being identical to entolysin A and the minor one identical to entolysin B, both originally purified from *P. entomophilia* L48^T^. From the full NMR analysis, we demonstrate that both entolysins, known to have the same mass, but different chromatographic retention profiles, nevertheless have identical planar structure, yet a different NMR fingerprint. We conclusively demonstrate that entolysin A and B only differ in the configuration of a single amino acid, and that this results in a different biological activity. This surprising find most likely indicates the presence of an ‘intermittently’ active E/C-domain and, in any case, shows that the NRPS assembly line not only allows for constitutional variations, but also configurational variations in the end product, thereby further expanding the molecular and functional space that can be covered.

## 2. Results

### 2.1. Dereplication of CLiPs Produced by Various P. entomophila Using NMR Fingerprint Matching

In a previous effort, we elucidated the planar structure and categorized the main CLiP metabolite produced by a variety of *Pseudomonas* strains extracted from the rhizosphere of the cocoyam plant [33,34]. Using MS and NMR, we showed that the main CLiP produced by *Pseudomonas entomophila* COR5 consists of a 3-hydroxy decanoic fatty acid linked to a 14-amino-acid peptide chain and a 5-residue macrocycle, leading to a 14:5 CLiP, following our previously introduced nomenclature [30], and with the sequence as shown in Table 1. As its primary sequence and overall amphipathicity are highly homologous to that of entolysin A and B purified and characterized from *P. entomophila* L48^T^ by Vallet-Gely et al. [35], it can be placed within the entolysin group. Even though the 14:5 CLiP from the COR5 strain has identical mass to entolysin A and B from *P. entomophila* L48, presumed identity with either of these CLiPs could not be conclusively established since the primary structures of the latter CLiPs remained ambiguous. Indeed, the inability of mass spectrometry to distinguish isobaric Leu from Ile leads to the attribution of four positions as Xle residues, an ambiguity which could not be resolved from bio-informatic analysis of the entolysin biosynthetic gene cluster (BGC). To definitively establish the structural identity while avoiding an extensive characterization effort, we proceeded to the early-stage dereplication approach using NMR fingerprint matching. Here, the pairwise comparison of the combined ^1^H and ^13^C chemical shift fingerprint from ^1^H-{^13^C} HSQC spectra for the CLiP backbones member of the same group is used to establish structural identity up to the level of the configuration of the individual amino acids [30].

Since no raw or tabulated NMR data have been reported for the entolysins isolated from *P. entomophila* L48, the latter was cultured to produce entolysin A and B in a 3:2 ratio. (Appendix A) In parallel, we also recultured *P. entomophila* COR5, which, in addition to the original entolysin-like CLiP, also produced a second homologue CLiP, in a 3:2 ratio (Appendix A), with identical mass to the major compound, as well as to both entolysins from the L48 strain. Without explicit resonance assignment of any NMR spectra, simple comparison of their (CHα) fingerprint (Figure 1a) shows that the main CLiP from the COR5 and L48 strains match perfectly, identifying both as entolysin A. Similarly, the minor CLiPs from both *P. entomophila* strains also provide a perfect match (Figure 1b), identifying both as entolysin B, as designated by Vallet-Gely et al. [35]. Importantly, the spectra of entolysin A and B clearly show distinct fingerprints—i.e., a clear mismatch (Figure 1c)—excluding the possibility for an erroneous A/B attribution when evaluating CLiPs from both strains, and further underlining their distinct molecular make up. Consequently, both *P. entomophilia* L48^T^ and *P. entomophila* COR5 produce entolysin A as the main compound and entolysin B as the minor compound. Going further, we extracted entolysin-like 14:5 CLiPs from other putative sources, notably, from *Pseudomonas* sp. COR6, *Pseudomonas* sp. COR16, *Pseudomonas* sp. COR17 and *Pseudomonas* sp. COW47. These are all isolated from cocoyam roots in Cameroon and taxonomically closely related to *P. entomophila* L48^T^ and *P. entomophila* COR5 [33]. For all these bacteria, we could again extract a major compound and a single minor compound in a 3:2 ratio, similar to that observed for COR5 and L48 (see Appendix A). Using NMR matching to the reference spectra of entolysin A, respectively, B from *P. entomophila* L48, we established that all these strains produce entolysin A as the major and entolysin B as the minor compound, with subsequent NMR analysis confirming the associated sequence. (Appendix A). Given this, from here on, the major and minor form isolated from bacterial growth will therefore be referred to as entolysin A and B, respectively, irrespective of the producing strain.

Since we previously elucidated the planar structure of entolysin A from the COR5 strain (Table 1), the fingerprint match also allowed for removing the Xle ambiguities in the original sequence proposal of Vallet-Gely et al. [35], thus establishing that a single isoleucine is present in entolysin A at the C-terminus. We next proceeded to elucidate the planar structure of the minor entolysin B compound, which had not been investigated by NMR before. Much to our surprise, the planar structure describing the type and position of amino acid residues of entolysin B proved identical to that of entolysin A, the same amino acid residues occurring at identical positions in both sequences, with both also featuring an identical N-terminal acyl chain. While this fits perfectly with entolysin A and B having the same molecular mass, it cannot explain the distinct (CHα) backbone fingerprint (Figure 1c) and different chromatographic retention times and invalidates the original proposal that both homologues differ by the position of a single isoleucine/leucine pair. A closer analysis of the chemical shift differences in the backbone fingerprint area (Appendix A) indicated differences in (CHα) chemical shifts to be localized towards the C-terminus and most pronounced for Ser13 and Leu12, as also indicated in Figure 1c. Based on previous assessment of the impact of D/L switches in NMR backbone fingerprints [30] and given the identical planar structure, we hypothesized that both entolysins differ in a D/L-configurational variation, even though such has, to the best of our knowledge, never been reported so far. To verify this hypothesis, the stereochemical makeup of entolysin A and B was in need of elucidation.

### 2.2. Entolysin A and B Differ in Configuration at Ser13

Using our previously proposed analysis workflow for the characterization of CLiPs, we first performed a configurational analysis of the individual amino acids by means of Marfey’s analysis. (Appendix A) This revealed the presence of 2x D-Leu, 2x L-Leu, 4x D-Glx, 1x L-Val, 2x D-Val and 1x L-Ile for both CLiPs (Table 1 and Appendix A). However, the minor entolysin B featured two D-Ser residues, while one L-Ser and one D-Ser were identified from the major homologue, entolysin A. Thus, it indeed appeared that both entolysins could differ in stereochemistry by a D-to-L switch in the configuration of a single serine at either position 10 or 13. While the latter residue experiences the largest H^α^C^α^ chemical shift difference when comparing the fingerprint of entolysins A and B, this does not provide unequivocal proof for an L-Ser13. Moreover, the occurrence of configurational heterogeneity for leucines (D:L 2:2) and valines (D:L 2:1) also produces a total of eight positional uncertainties in both entolysin sequences when it comes to D/L-configuration. We thus resorted to total synthesis of a reduced library of entolysin diastereoisomeric sequences, together with their NMR fingerprinting against the natural analogues, to establish and revise the stereochemistry of various CLiPs.

Based on the leucine and valine configurational heterogeneity, up to 18 different configurational combinations need to be considered for entolysin B, which doubles to 36 for entolysin A, as here the serines occur in a 1:1 D:L ratio, as well. To reduce the synthesis effort, this library of possible diastereoisomers was reduced to a more manageable size through bioinformatic analysis of the biosynthetic gene clusters.

The NRPS assembly line producing entolysins in *P. entomophila* COR5 is encoded in three genes (*EtlA*, *EtlB* and *EtlC*) that provide, respectively, two, eight and four CAT-modules. Assuming co-linearity, the predictive analysis of the A-domains from the associated NRPS proposes a Leu–Glu–Gln–Val–Leu–Gln–Val–Leu–Gln–Ser–Val–Leu–Ser–Ile tetradecapeptide sequence (Table 1), which represents a perfect match with the sequence established using spectroscopic methods (see above). Next, the nature of the C-domains was evaluated to identify ^L^C_L_- and E/C-domains. The initial C_start_-domain responsible for the incorporation of the fatty acid Is followed by ten E/C-domains. Each of these is responsible for the condensation of the newly recruited L-amino acid to the growing peptide chain, along with L-to-D epimerization of the preceding residue in the sequence. Thus, residues 1 to 10 are all predicted to be D-configured (Table 1). This is followed by two ^L^C_L_-type domains, which lack the epimerization functionality, thus causing the L-configuration Val11 and Leu12 to be retained. The C-terminal module features an E/C-domain, suggestive of both the epimerization of the amino acid at position 13 and the incorporation of an L-amino acid as the final residue before transfer to a tandem of thioesterase (TE) domains for cyclisation with concomitant product release. The predicted configurations associated with positions 1 to 13 match those proposed earlier from the bioinformatic analysis of the *P*. *entomophila* L48^T^ BGC [35]. While helpful, one is reminded of the merely predictive nature of the amino acid configurations, as a 3:1 D:L ratio is proposed for the four leucine residues, whereas a 2:2 D:L ratio is observed from Marfey’s analysis. As mentioned earlier, the possibility of inactive E/C-domains implies that one can only establish with certainty that amino acids introduced at the position (n − 1) preceding an ^L^C_L_-domain in module n are L-configured (Table 1). Thus, the fact that the 1^2t^h module has a ^L^C_L_-domain—which is incapable of epimerization—implies that the valine previously incorporated by module 11 must be L-Val. The position of the only L-Val now firmly established, the position of both D-Val residues automatically follows, and the number of sequences to consider is reduced by a factor of three. Likewise, the C-domain of module 13 is also an ^L^C_L_, allowing for positioning one L-Leu with certainty at position 12, leaving three, respectively, six possible ways to distribute two D-Leu and one L-Leu in the sequence for entolysin A/B. At this point, no further elements are available to securely reduce the possibilities any further.

In order to prioritize the remaining sequences in an effort to manage the synthesis effort, we performed additional phylogenetic analysis of currently available E/C-classified domains, incorporating experimental information on epi-inactive E/C-domains as included in [36] (see Appendix A for more details; Appendix A). The second C-domain of EtlA clusters with a clade of C-domains present in the NRPS that synthesize xantholysin and with Viscosin group members (WLIP, viscosin, viscosinamide, massetolide, pseudodesmin, pseudophomin), as well as poaeamide and orfamide (Appendix A). In all cases, the available configurational analysis of the NRPS products indicates these E/C-domains lack epimerization activity. Based on the pronounced domain homology of the E/C-domain in module 2 of the *etlA* gene, we therefore proposed that this E/C-domain is most likely also epi-inactive and would therefore place the second L-Leu reported from Marfey’s analysis at the N-terminus. Consequently, for entolysin B, this prioritizes two sequences for synthesis, differing only in the configuration of Ser10 and Ser13 (Table 1). The 3-hydroxy fatty acid tail was initially assumed to be (*R*)-configured, as is generally observed for *Pseudomonas* CLiPs and confirmed later on through NMR fingerprinting, which is sensitive to the fatty acid configuration, as well [30].

Taken together, we synthesized and purified three 14:5-CLiPs, with D-Ser10/L-Ser13 (**1**), L-Ser10/D-Ser13 (**2**) or D-Ser10/D-Ser13 (**3**), using our previously established total synthesis approach (see Appendix A) [37]. Collected in Table 1, the latter (**3**) should match with entolysin B, while either (**1**) or (**2**) should match with the entolysin A.

Figure 2 shows the overlay of the (CHα) fingerprint region of the ^1^H-{^13^C} HSQC in DMF-*d7* of each synthesized (14:5) sequence with that of natural entolysin A and B from *P. entomophila* COR5, all recorded under otherwise identical conditions. A straightforward visual assessment clearly indicates that the fingerprint of the L-Ser13 (**1**) variant displays a quasi-perfect match with entolysin A (Figure 2a), while the fingerprint of the L-Ser10 (**2**) variant is clearly mismatched (Figure 2b), thus clearly indicating Ser 10 and Ser13 are D- and L-configured, respectively. As can be expected, the fingerprint of the D-Ser10/D-Ser13 (**3**) homologue is also mismatched (Figure 2c). Considering these three variants against entolysin B (Figure 2d–f), an excellent match is only observed with D-Ser10/D-Ser13 (**3**) (Figure 2f), clear mismatches appearing with the other L-Ser-containing homologues (Figure 2d,e). The match of entolysin A, respectively, entolysin B with L-Ser13 (**1**) and D-Ser10/D-Ser13 (**3**) also settles the configuration of L-Leu1, inferred from C-domain analysis (see above). The results obtained here definitively prove, therefore, that the entolysin NRPS systems in *P. entomophila* L48^T^ and COR5 produce diastereomeric lipopeptides, only differing in the configuration of the penultimate Ser13 residue.

All data together establish that the stereochemistry of the main entolysin A corresponds to that of the l-Ser13 (**1**) sequence 3R-OH C10:0–L-Leu–D-Glu–D-Gln–D-Val–D-Leu–D-Gln–D-Val–D-Leu–D-Gln–D-Ser–L-Val–L-Leu–L-Ser–L-Ile, while the minor entolysin B corresponds to that of the d-Ser10/ d-Ser13 (**3**) sequence 3R-OH C10:0–L-Leu–D-Glu–D-Gln–D-Val–D-Leu–D-Gln–D-Val–D-Leu–D-Gln–D-Ser–L-Val–L-Leu–D-Ser–L-Ile, whereby the underlined amino acids indicate those that are part of the macrocycle. Since all analyzed entolysin producers biosynthesize entolysin A and B, the presence of a stereochemical heterogeneity between the major and minor seems to be a common feature of entolysin systems. So far, this observation is unique to the entolysins.

### 2.3. Intermittently Epi-Inactive E/C-Domain as Source of Configurational Variability

Since both entolysin homologues are produced by a single tripartite NRPS system and this recruits L-amino acids from the primary metabolism, the stereochemical heterogeneity at Ser13 most likely originates because the E/C-domain of module 14 is intermittently active in its epimerization activity. Conventional C-domains, only capable of condensing L-amino acids (i.e., ^L^C_L_-domains) into the growing peptide chain and devoid of epimerization functionality, feature an active site with a conserved histidine-containing H**H**xxxDG motif, in which the second histidine (**H**) is crucial for catalysis [15,38]. The dual E/C-domains present in *Pseudomonas* CLiPs carry an equivalent H**H**xxxDH motif, with an extra histidine (or less frequently, alanine or asparagine) substituting for the glycine [15,39]. In addition, the dual E/C-domains invariably exhibit a second histidine sequence motif, with the proposed consensus sequence HH[I/L]xxxxGD located close to the N-terminus of the domain [23,38]. In effect, the presence of this second motif allows for identifying the presence of E/C-domains and, therefore, predicts the presence of D-amino acids in the sequence. As noted above, experimental verification of the configuration of amino acids in *Pseudomonas* CLiPs has shown that the epimerization by these E/C-domains does not always occur. While several other motifs have been tentatively linked to the inability of E/C-domains to perform epimerization, there is no clear consensus on this [23]. Indeed, this is hampered by substantial variation in the consensus HH[I/L]xxxxGD second histidine sequence motif in *Pseudomonas* species, both for those epi-active and those epi-inactive (Figure 3).

Sequence alignment of 74 inactive E/C-type domains and 394 active E/C-type domains reveals no striking difference in the secondary motifs associated with epimerization activity or inactivity. However, in comparing all these E/C-type domains, we noticed a difference in the flanking sequence just before the secondary histidine motif of EtlC-C14. Since the E/C-domain of a particular module epimerizes the amino acid that is incorporated by the preceding module, EtlC-C14 determines the configuration of Ser13 in the entolysin structure and, from the above, appears to be intermittently epi-active. More specifically, a conserved leucine residue is replaced by an extra histidine in this E/C-domain (Figure 4; light green box) in front of the secondary motif (Figure 4; yellow box). This is the case for both known entolysin producers (strains COR5 and L48), as well as two candidate entolysin producers (*P. entomophila* strains Small and 23S). These four strains represent independent isolates from different environments: Small from soil (Arkansas, USA), 23S from agricultural rhizospheric soil (Quebec, Canada), L48 from fly (Guadeloupe) and COR5 from cocoyam roots (Cameroon). This further support the relevance of the observed differences, confined to *P. entomophila.* The NRPS systems of these four strains are closely related (about 95% pairwise amino acid identities). The atypical histidine is not present in the 10 other E/C-type domains (C2 through C11) of the four different isolates and is absent from all other E/C-type domains (epi-active or epi-inactive) of *Pseudomonas* (sequences are available at the RhizoCLiP website), including those of several xantholysin-producing species (*P. maumuensis*, *P. mosselii*, *P. muyukensis*, *P. soli*, *P. xantholysinogenes*). These phylogenetical relatives of *P. entomophila* carry NRPS systems that display the highest domain homologies outside the entolysin producers (Appendix A). This divergence of the secondary motif is also notable in partial EtlC sequences of candidate entolysin producers (*P. entomophila* Au-Pse2, GenBank JAKSEJ010000001; *P. entomophila* MWU13-3659, GenBank JAMSHW010000004). We also could not identify such a His-supplemented motif in other *Pseudomonas* E/C-type domains that have not yet been experimentally verified.

Given the conservation of the neighboring His in the signature motifs of E/C-type domains, we hypothesize that this extra His has an effect on the intrinsic epimerase activity. As far as we know, no 3D structure of an E/C-domain has been determined yet. This could potentially yield relevant information about the positioning of the extra His relative to the putative active site.

### 2.4. Biological Relevance of Configurational Heterogeneity

Reports on the biological activities of entolysin are scarce, with limited data available for entolysin A, while none is available for the minor compound, entolysin B. Upon their extraction from *P. entomophilia* L48, entolysin A and B were proposed to be responsible for its producers’ pathogenicity against insects [35]. However, while the study revealed a clear role for entolysin in swarming motility of its producing organism, it was not the main virulence factor against *Drosophila melanogaster*. This study also showed a mutant unable to produce entolysin had similar biocontrol activity as the wild-type strain in a cucumber-*Pythium ultimum* pathosystem as the wild-type strain *P. entomophilia* L48, suggesting that entolysin is not involved in *Pythium* biocontrol [35]. Later on, however, it was demonstrated that entolysin is capable of suppressing the cocoyam root rot pathogen *Pythium myriotylum* in a dose-dependent manner and causes hyphal leakage [23]. In addition, crude entolysin extracts can control the rice blast fungus *Pyricularia oryzae* by direct antagonism and via induced systemic resistance [41]. Interestingly, the entolysins are produced in a 3:2 ratio by the various *P*. *entomophila* strains, as opposed to other *Pseudomonas* CLiPs, where a single major compound largely dominates over the production of minor forms. The unique observation that entolysin A and B only differ in their stereochemistry, while being produced by a single NRPS system, raises questions on their biological relevance. To address this, we performed antifungal assays on spores of *Botrytis cinerea* R16 and mycelium of *P. oryzae* VT5M1.

Both compounds were able to permeabilize the fungal spore membranes of *P. oryzae* VT5M1, starting at 16 μM. Moreover, entolysin B had a significantly higher permeabilizing activity compared to entolysin A (Figure 5A,B). This was also found with mycelium of *B. cinerea* R16, where entolysin B was more active than entolysin A, and both of them showed permeabilizing activity, starting from 32 μM (Figure 5C). However, entolysin B was less active than entolysin A when tested on mycelium of *P. oryzae* VT5M1 at concentrations below 32 μM (Figure 5D), but showed equal activity at 64 µM. Taken together, these results indicate a higher antifungal activity of entolysin B compared with entolysin A against *B. cinerea* and *P. oryzae* spores and *B. cinerea* mycelium. The (10:8) CLiP orfamide B, not active in our assays, was used as a negative control.

A checkerboard assay was used to study a possible synergistic activity between entolysin A and entolysin B (Figure 6A) [42]. At a 1:1 ratio, no synergy was observed at any of the concentrations (Figure 6B) or time points tested, whereas it was further confirmed that entolysin B is more active than entolysin A towards spores of *B. cinerea* R16.

## 3. Discussion

It is often observed that flexibility in the A- and C_start_-domains leads to a variety of CLiP structures that can be produced by a single pseudomonad NRPS system by introducing an amino acid or fatty acid moiety that slightly varies from that normally introduced in the structure of the major compound. This leads to the production of multiple CLiP homologues by a single bacterium, even though there is a single NRPS system responsible for the biosynthesis. In other *Pseudomonas* bacteria hosting similar BGC’s, the associated NRPS may then encode the production of an already described minor compound as major, leading to confusion in the nomenclature (e.g., both viscosin and massetolide production in a single *Pseudomonas* species) [22]. To aid in the categorization of *Pseudomonas* CLiPs, we assemble them into different groups according to the total length of the peptide sequence and the size of the macrocycle, i.e., its (l:m) tag [30]. In addition to variability in amino acid or fatty acid identity, within a particular (l:m) group, differences can also occur in amino acid configurations. However, these compounds are typically produced by genetically distinct NRPS systems. For example, viscosin and WLIP both belong to the Viscosin (9:7) group and differ only in the configuration of Leu5, being L-Leu5 in viscosin and D-Leu5 in WLIP. Although the bioinformatic analysis of the NRPS systems of both molecules shows the presence of an E/C-domain in module 6, this domain 6 is epi-inactive in the case of viscosin (thereby introducing a L-Leu5), while it is epi-active in the case of WLIP (which thus features a D-Leu5). Up until now, it was generally assumed that the manifestation by a specific E/C-domain of epimerization activity is a binary feature; it is either fully able or fully unable to epimerize an amino acid. In other words, there is no production of WLIP in case of a viscosin-producing NRPS system, or vice versa. Any minor compounds were, thus, thought to be uniquely the result of A- or C_start_-domain flexibility. However, in this report we show, by the characterization of entolysin A and B, that this is not always the case. More specifically, using solid-phase peptide synthesis of a small library of entolysin sequences and NMR fingerprint matching, we established that entolysin A and B are diastereomeric homologues, differing only in the configuration of Ser13. This is a remarkable observation that, to the best of our knowledge, represents the first report of configurational homologues produced by a single NRPS system in a *Pseudomonas* strain. Moreover, this atypical feature is elaborated by multiple strains of the *P. entomophila* species, originating from different biological and geographical sources. Finally, whereas minor compounds originating from A- or C_start_-domain flexibility are generally less than 5% of the major compound, the ratio between entolysin B and entolysin A is much higher (3:2).

Bioinformatic analysis reveals the presence of an E/C-domain in module 14 in the entolysin-producing *P. entomophila* L48^T^ and COR5, predicting the presence of D-Ser13. While this is indeed what is experimentally found for entolysin B, using peptide synthesis and NMR fingerprint matching, entolysin A was identified as a diastereomeric homologue featuring an L-Ser13. Therefore, it appears that the minor compound, entolysin B, is not produced as a result of flexibility in a A- or C_start_-domain, as previously assumed, but rather due to an intermittently epi-active E/C-domain in module 14 of its NRPS. Sequence alignment of the E/C-domain of module 14 with those of *Pseudomonas* E/C-type domains with experimentally established behavior with respect to epimerization revealed that this deviant behavior coincides with the presence of an extra histidine in the secondary histidine motif characteristic for the E/C-domain. Further research will be required to establish a causal relationship, e.g., by the construction of the presumed loss/gain of function mutants, such as His → Leu in EtlC-C14 of *P. entomophila* COR5 or L48 and/or Leu → His in a xantholysin producer with a strong homology (XtlC). However, this goes beyond the scope of the current article.

Antimicrobial activity of CLiPs is believed to be due to permeabilization of cell membranes, possibly through pore formation. This results in destabilization and disruption of the membrane, leading to cell death by causing leakage of cellular contents [43,44]. Propidium iodide (PI) staining is frequently used to evaluate the integrity of the cell membrane. The assay is predicated on the ability of PI to enter cells whose membranes have been disrupted and to intercalate into DNA, which results in an increase in the fluorescence signal [45].

In our study, we investigated the antifungal activity of entolysin A and entolysin B with the PI staining assay. The minor compound, entolysin B, was clearly more active against spores and mycelium of *B. cinerea* R16 and spores of *P. oryzae* VT5M1 compared with the major entolysin A (Figure 5). This is very intriguing since both compounds only differ in the configuration of a single amino acid in the macrocycle. Despite the difference in activity, we did not observe any synergy for entolysin A and B, suggesting that they may have similar modes of action and target the same components in the fungal membrane, the D/L-configuration possibly modulating their membrane partitioning or membrane perturbation potential.

## 4. Materials and Methods

### 4.1. General Methods

The 2-chlorotrityl chloride-linked polystyrene resin (2-CTC resin) (1.60 mmol/g), all l-amino acids, Fmoc-d-Val-OH, Fmoc-d-Gln(Trt)-OH, HBTU, TFA, DIC and Allyloxycarbonyl succinimidyl ester (Alloc-OSu) were purchased from Iris Biotech GmbH. Oxymapure, HOBt, dry DIPEA, piperidine, phenylsilane, triisopropylsilane, tetrakistriphenylphosphine palladium (0), (cod)Ru(2-methylallyl)2, Nα-(2,4-Dinitro-5-Fluorophenyl)-L-alaninamide, (*R*)-BINAP, L-amino standards and allylbromide were obtained from Sigma Aldrich (Saint Louis, MO, USA), and HATU and the D-amino acids, including Fmoc-d-Leu-OH, Fmoc-d-Glu(tBu)-OH, Fmoc-d-allo-Thr(OH)-OH and Fmoc-d-Ser(tBu)-OH, were ordered from Chem-Impex International (Wood Dale, IL, USA). Octanoyl chloride, DMAP and 1-methylimidazole were purchased from Acros Organics (Geel, Belgium), and hydrobromic acid was obtained from Janssen Chimica (Beerse, Belgium). Peptide-grade NMP and DMF obtained from Biosolve (Valkenswaard, The Netherlands) were used throughout the on-resin synthesis and washing of the resin. Dry THF, DCM, DMF, MeOH, pyridine and all other solvents were purchased from Acros Organics and used without further purification or drying.

#### 4.1.1. LC-MS Analysis

LC-MS analysis was performed on an Agilent (Santa Clara, CA, USA) 1100 Series HPLC with an ESI detector type VL, equipped with a Kinetex column (C18, 150 Å, 4.60 mm, 5 µm particle size) with a flow rate of 1.5 mL/min. Two different solvent systems were used, either 5 mM NH_4_OAc in H_2_O (A) and CH_3_CN (B) or 0.1% CF_3_COOH in H_2_O (A) and CH_3_CN (B). Optimized different gradients were used for each CLiP.

#### 4.1.2. High-Resolution Mass Spectrometry MS(TOF)

High-resolution mass spectra were recorded on an Agilent 6220A time-of-flight mass spectrometer, equipped with an Agilent ESI/APCI multimode source. The ionization mode was set to APCI (atmospheric pressure chemical ionization), while the mass spectra were acquired in the 4 GHz high-resolution mode, with a mass range set to 3200 Da.

#### 4.1.3. HPLC

Semi-preparative purification was performed on an Agilent 218 solvent delivery system with a UV-VIS dual wavelength detector, using a Phenomenex column (AXIA packed Luna C18(2), 250 Å, 21.2 mm, 5 µm particle size) with a flow rate of 17.5 mL/min and the following solvent systems: H_2_O containing 0.1% TFA (A) and CH_3_CN (B).

Preparative purification of the crude peptides via RP-preparative-HPLC was performed on a Gilson PLC 2250 instrument, with a Waters Millipore Corporation DeltaPak C18 PrepPak cartridge (pore size 100 Å and spherical silica of 15 µM) connected to a multiwavelength detector. The solvents used for the measuring of samples are H_2_O (A) and MeCN (B), both containing 0.1% CF_3_COOH, at a flow rate of 65 mL min^−1^.

### 4.2. Synthesis of Building Blocks

The building blocks necessary for the synthesis of the entolysin analogues, including Fmoc-d-Ser(OH)-OAll, Fmoc-l-Ser(OH)-OAll, Alloc-l-Ile and the lipid tail ((*R*)-3-(tert-butyldimethylsilyloxy)decanoic acid), were synthesized according to the methods described in the literature [37]. The enantiomeric excess of the lipid tail was determined using chiral HPLC analysis run on a Daicel Chiracel ODH column (250 × 4.6 mm), with an isocratic elution of hexane/EtOH (97/3) over 30 min and a flow rate of 1 mL/min. An enantiomeric excess of 99.0% of the (*R*)-enantiomer was established after the asymmetric hydrogenation step. The obtained NMR spectral data were in accordance with earlier reported data.

### 4.3. Solid-Phase Peptide Synthesis

#### 4.3.1. Automated Peptide Synthesis

Automated peptide synthesis was performed in plastic reaction vessels equipped with Teflon frits (MultiSyn Tech GmbH (Witten, Germany)). The automated peptide synthesis was executed with an INTAVIS (multipepRsi) Multiple Peptide Synthesizer Robot, equipped with a vortex unit, at room temperature. The solid-phase peptide synthesis made use of the ubiquitous Fmoc/tBu protecting strategy, with OxymaPure/DIC as coupling reagents. The Fmoc deprotection step was performed by treatment with a solution of 40% piperidine in DMF (*v*/*v*) for 4 min and repeated for another 12 min. The coupling step was carried out on the resin using Fmoc-protected amino acids (5 equiv.) dissolved in DMF in the presence of 0.5 M OxymaPure (5 equiv.) and 0.5 M DIC (5 equiv.) in the presence of 0.05eq DIPEA for 40 min at room temperature. The coupling step was performed twice to ensure complete coupling of the amino acids.

#### 4.3.2. Manual Peptide Synthesis

For a manual removal of the Fmoc group, the resin was typically treated with 20% piperidine in DMF (*v*/*v*) (10 mL/g resin) for 5 min; then, the reaction mixture was filtered and washed with DMF. The reaction was subsequently repeated twice for 10 and 20 min. Next, the resin was washed with DMF, MeOH and DCM.

For manual coupling of a building block, 5 equiv. OxymaPure in DMF (0.5 M) was added to a solution of 5 equiv. of Fmoc-protected amino acid in DMF (0.5 M) and shaken for 1 min. Then 5 equiv. DIC (0.5 M) in the presence of 0.05eq DIPEA in NMP was added to the previous solution and shaken for 2 min before adding it to the resin. After 1 h of shaking, the reagent was filtered, and the resin was washed. The outcome of the reaction can be monitored with a small-scale cleavage and subsequent LC-MS analysis on the obtained peptide.

For manual coupling of the lipid tail, 5 equiv. of OxymaPure in DMF (0.5 M) was added in 7 equiv. of the fatty acid (0.5 M), then 5 equiv. DIC (0.5 M in NMP) in the presence of 0.05 equiv. DIPEA was subsequently added. The mixture was sonicated and added to the resin. After 2 h of shaking, the reagent was filtered, and the resin was washed. The outcome was monitored with a small-scale cleavage and subsequent LC-MS analysis on the obtained peptide.

#### 4.3.3. Peptide Cleavage

Intermediate compounds were analyzed by subjecting a small fraction of the peptidyl resin to an acidic cleavage. The released peptide was then analyzed by LC-MS. The following small-scale cleavage conditions with TFA were typically applied: 1 mg of peptidyl resin was brought in a plastic reactor vessel equipped with a Teflon frit. Then, 1 mL of a mixture of TFA: TIS: H_2_O (95:2.5:2.5% *v*/*v*/*v*) was added, and the vessel was shaken for 30 min. The resin was washed with DCM (1 mL), and the combined filtrates were dried using an argon stream to obtain the crude peptide. Subsequently, the peptides were precipitated in cold MTBE, while the protecting group debris remained dissolved.

The final cleavage and deprotection step were performed as follows: a solution of TFA:TIS:H_2_O (95:2.5:2.5% *v*/*v*/*v*); 10 mL/g resin) was added to the resin and shaken for 40 min. The mixture was filtered, and the resin was washed three times with 1 mL TFA. The filtrate was collected in a Falcon^®^ tube. The cleavage reaction was repeated twice, and the combined filtrates were dried (using Ar or N_2_) to obtain the crude peptide. Thereafter, cold MTBE was added to precipitate the peptidic compounds present in the crude mixture. Precipitation of all peptidic compounds was ensured by 8 min of centrifugation (10,000 r.p.m). The supernatant, only containing the protecting group debris, was decanted afterwards. This precipitation step was repeated at least three times. The remaining MTBE was evaporated with a gentle argon stream.

### 4.4. Production and Extraction of Natural Cyclic Lipodepsipeptides

A streak of bacterial cells of *Pseudomonas* sp. was transferred into 4 × 5 mL King’s Broth medium (Difco Laboratories, Sparks, MD, USA) and grown for 1 day under equilibrated conditions (29 °C, 140 rpm shaking frequency). Afterwards, each of the 5 mL cultures were transferred into a 2 L Erlenmeyer flask containing 400 mL of KB or M9 minimal medium [40]. Following an additional 24 h cultivation under minimal conditions (28 °C, 150 rpm shaking frequency), the culture was centrifuged to separate the supernatant from the cells.

The supernatant was acidified towards a pH of 3 with use of a 2 M HCl solution and was kept at 5 °C overnight. Subsequently, the supernatant was once again subjected to centrifugation to collect the precipitated peptide products. The resulting crude mixture was dissolved in a minimal amount of acetonitrile/water (75/25) prior to purification. An optimal elution gradient of 25/75 → 0/100 H_2_O/CH_3_CN was applied over 20 min at a flow rate of 17.5 mL min^−1^, while the column temperature was kept at 35 °C.

### 4.5. Characterization of Compounds Using NMR Spectroscopy

NMR characterization of intermediate products: ^1^H NMR and ^13^C NMR were recorded in CDCl_3_ on a Bruker Avance spectrometer equipped with a 5 mm BBO probe and operating at 300 MHz and 75.77 MHz, respectively.

NMR measurements of the final compounds were performed on either a Bruker Avance III spectrometer operating at a respective ^1^H and ^13^C frequency of 500.13 MHz and 125.76 MHz equipped with a BBI-Z probe or a Bruker Avance II spectrometer operating at a respective ^1^H and ^13^C frequency of 700.13 MHz and 176.05 MHz and equipped with either a ^1^H, ^13^C, ^15^N TXI-Z probe or 5 mm Prodigy TCI probe. The sample temperature was set to 298.0 K in all cases. High-precision 5 mm NMR tubes (Norell Inc., Morganton, NC, USA) were used for all NMR experiments.

Full assignment of the peptides was achieved through liquid-state NMR spectroscopy using 1D ^1^H, 2D ^1^H-^1^H COSY, ^1^H-^1^H TOCSY with 90 ms spinlock, ^1^H-^1^H off-resonance ROESY with a mixing time of 200 ms and gradient-enhanced ^1^H-^13^C HSQC and ^1^H-^13^C HMBC optimized for a ^n^J_CH_ coupling of 6.5 Hz. The spectral width was set to 14 ppm in the ^1^H dimension and 90 ppm (gHSQC) or 200 ppm (gHMBC) along the ^13^C dimension, and, generally, 2048 data points were sampled in the direct dimension and 512 data points in the indirect dimension. Standard pulse sequences available in the Bruker library were used throughout with excitation sculpting when applicable. For 2D processing, the spectra were zero filled to a 2048 × 2048 real data matrix, and all spectra were multiplied with a squared cosine bell function in both dimensions or a sine bell in the direct dimension for gHMBC before Fourier transformation. Assignment of all compounds was performed manually using the Bruker Topspin 4.1 software.

### 4.6. Permeabilization Assays Using Fungal Spores and Mycelium

For spore permeabilization assays, *Botrytis cinerea* R16 [46] and *Pyricularia oryzae* VT5M1 [47] were grown on, respectively, potato dextrose agar and oatmeal agar plates incubated at 28 °C for 7 days. Spores were collected by washing the mycelium using sterilized phosphate buffered saline (PBS) (pH 7.3). All spores were counted using a Thoma chamber and further adjusted to 5 × 10^4^ spores/mL. Thereafter, propidium iodide (PI—Invitrogen, ThermoFisher Scientific, Waltham, MA, USA) stock (1 mg/mL dissolved in water) was added to the spore suspension to obtain 10 µg/mL. A volume of 50 μL of spore suspension supplemented with 10 μg/mL PI was transferred in each well of a black 96-well plate (Greiner, black polystyrene, flat bottom). Each stock compound (entolysin A, entolysin B and orfamide B) dissolved in DMSO was diluted twice; the desired final concentration in sterilized PBS and 50 μL was inoculated in each well to obtain 5 µg/mL PI with a final concentration (0.5, 1, 2, 4, 8, 16, 32 and 64 μM) of each tested compound. A spore suspension of 50 μL (with 10 µg/mL PI) inoculated with 50 μL of sterilized PBS containing the same amount of DMSO used in the serial dilutions was treated as the control. The plate was incubated at 28 °C and measured (excitation 540 nm, emission 620 nm, 4 measurements per well, 8 flashes) with a microplate reader (Infinite 200 Pro M Plex; Tecan, Kawasaki, Japan) every one hour for a maximum of five hours.

The mycelium permeabilization assay was performed similarly, as described before [43]. *B. cinerea* R16 and *P. oryzae* VT5M1 were pre-cultured for 7 days on complete medium (CM) agar plates [48] at 28 °C. A 1.0 cm mycelium plug from each fungus was cut and blended in 50 mL CM broth for 30 s at a maximum speed, then the fusion was transferred into a 100 mL flask and incubated on a rotary shaker for 24 h at 28 °C, 150 rpm. After 24 h, the mycelium was re-blended, and 10 mL was transferred into a new flask containing 40 mL of CM broth and incubated under the same conditions for 24 h. A volume of 10 mL of each culture was filtered through a 40 μm cell strainer (SPL Life Sciences, Pochon, Republic of Korea). The mycelium was then washed with sterilized PBS (pH 7.3) and resuspended in 50 mL of sterilized PBS. Then, a stock solution of 1 mg/mL propidium iodide (PI) in water was added to the mycelium suspension to obtain 10 μg/mL. A volume of 50 μL of mycelium suspension supplemented with PI was transferred in each well of a black 96-well plate (Greiner, black polystyrene, flat bottom) and further processed as described above.

### 4.7. Checkerboard Analysis

A possible interaction between entolysin A and entolysin B against spores of *B. cinerea* R16 was investigated using the checkerboard method, with slight modifications [42]. The spores were collected from 7-day-old *B. cinerea* R16 on PDA plates. The fluorescence of PI staining of *B. cinerea* R16 was monitored in the presence of entolysin A at concentrations of 0, 0.5, 1, 2, 4, 8, 16, 32 and 64 μM in combination with entolysin B at concentrations of 0, 1, 2, 4, 8, 16, 32 and 64 μM. The experiment was performed in triplicate. Fluorescence values for entolysin A, entolysin B and their combinations were compared by two-way ANOVA analysis.

## Figures and Tables

**Figure 1 ijms-24-14302-f001:**
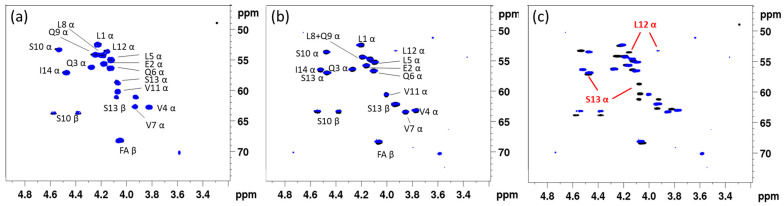
Dereplication of lipopeptides using ^1^H-{^13^C} HSQC (CH*α*) fingerprints. (**a**) Comparison of entolysin A (14:5) extracted from *P. entomophila* L48^T^ (blue) with that of entolysin A extracted from *Pseudomonas entomophila* COR5 (black). (**b**) Comparison of entolysin B (14:5) extracted from *P. entomophila* L48^T^ (blue) with that of entolysin B extracted from *P. entomophila* COR5 (black). While not required for fingerprint matching, amino acid assignments of A and B are indicated. (**c**) Comparison of entolysin A (black) and entolysin B (blue), both extracted from *P. entomophila* COR5, highlighting an overall different fingerprint. The largest differences, occurring at Leu12 and Ser13, are indicated. All spectra were recorded in DMF-*d7* at 500 MHz and 298 K.

**Figure 2 ijms-24-14302-f002:**
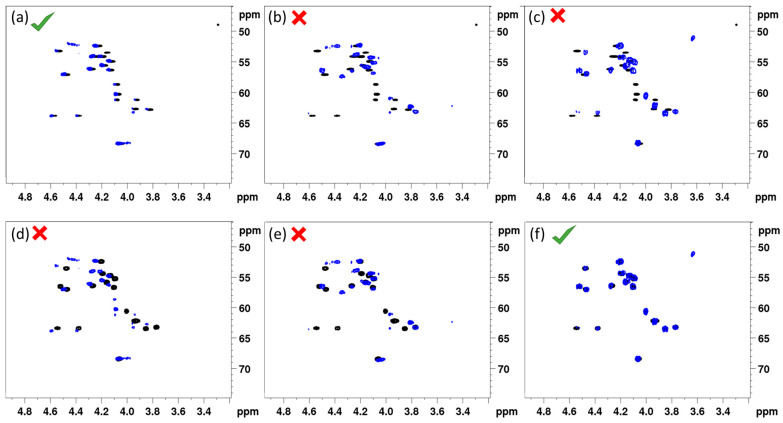
Comparison of the ^1^H-{^13^C} HSQC (CHα) fingerprints of the three synthetic (14:5) entolysin variants (blue) with that of entolysin A (horizontal, (**a**–**c**) and entolysin B (horizontal, **d**–**f**) produced by *P*. entomophila COR5 (black) recorded in DMF-*d7* at 700 MHz, 298 K. Matches shown are with L-Ser13 (**1**) (left, (**a**,**d**)), L-Ser10 (**2**) (middle, (**b**,**e**)) and D-Ser10-D-Ser13 (right, (**c**,**f**)), respectively.

**Figure 3 ijms-24-14302-f003:**
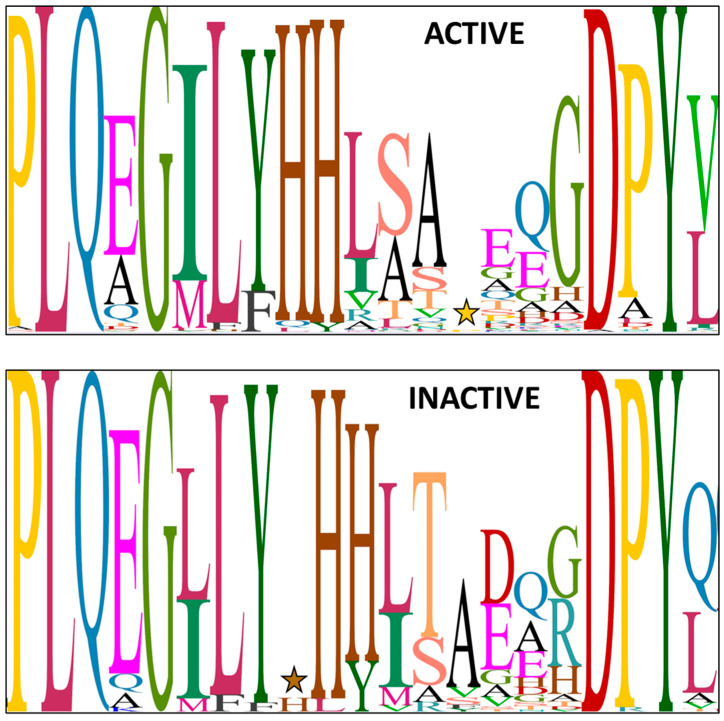
Sequence logo for the N-terminal secondary histidine motif present in E/C-type domains of *Pseudomonas* lipopeptide NRPS systems, excluding *P. entomophila* NRPS systems. Herein, the relative proportion of a particular amino acid at a particular position is represented by the size (height) of the amino acid. The sequence logo for active, respectively, inactive E/C-domains is based on 394 and 74 sequences, respectively, with experimentally proven outcome with regards to epimerization activity. A star marks the location of an apparent amino acid insertion (histidine or proline) occurring in some E/C-domains. More specifically, ACTIVE panel (top): the ‘gap’ is due to an extra residue (proline) in some WLIP synthetases (WlmC, C9; COW40, COR54); INACTIVE panel (bottom): the ‘gap’ (small H) is due to an extra residue (histidine) in JesB/jessenipeptin (C8) and in FusA/fuscopeptin (C4) [40].

**Figure 4 ijms-24-14302-f004:**
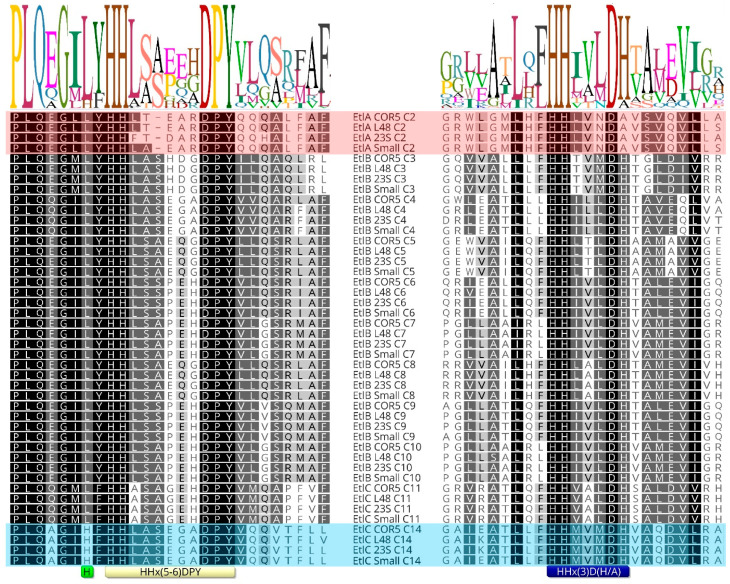
Sequence logo for the primary and secondary histidine motifs present in E/C-type domains of *P. entomophila* lipopeptide NRPS systems. Sequences aligned on the right cover the common C-domain motif (consensus motif in blue box). The secondary histidine motif (yellow box) is indicated for the left sequence alignment. The extent of sequence conservation is visualized by differential grey shading of amino acid residues. Sequences of the epi-inactive C2-domains (light red background) and the intermittently active C14-domains of entolysin producers COR5 and L48 (light blue background) are highlighted. The position of an additional histidine adjacent to the C14 secondary histidine motif is marked by a green box. The corresponding NRPS sequences from potential entolysin producers *P. entomophila* Small (GenBank CP070982) and 23S (GenBank CP063832) are included for comparison. The extent of sequence conservation is visualized by differential grey shading of amino acid residues and shows similarity based on the Blosum62 score matrix: black (100%), dark grey (80–99%), light grey (60–79%), white (<60%).

**Figure 5 ijms-24-14302-f005:**
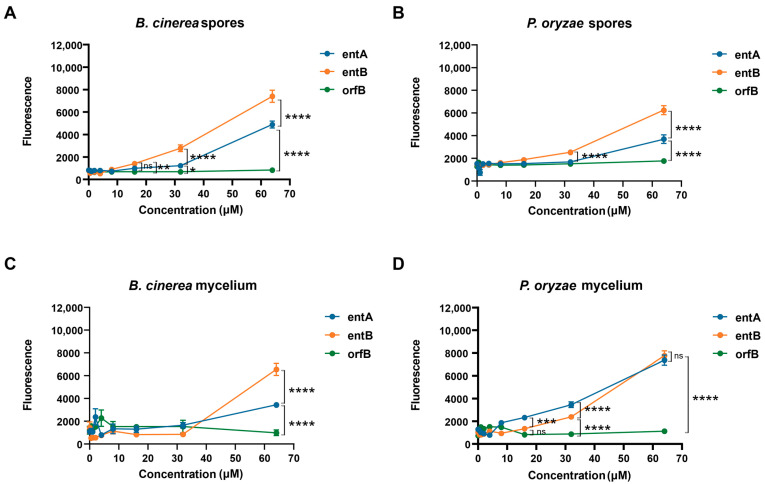
Entolysin A and entolysin B cause membrane leakage of fungal mycelium and spores. Entolysin A (entA), entolysin B (entB) and orfamide B (orfB) activity against (**A**) *B. cinerea* spores, (**B**) *P*. *oryzae* spores, (**C**) B. cinerea mycelium, (**D**) *P. oryzae* mycelium. The fluorescence, shown in the figure, was measured 2 h after treatment. The bars indicate the standard error (*n* = 8). Entolysin A, entolysin B and orfamide B were compared by two-way ANOVA analysis. Significant differences at *p* > 0.05, *p* ≤ 0.05, *p* ≤ 0.01, *p* ≤ 0.001 and *p* ≤ 0.0001 were marked as (ns), single (*), double (**), triple (***) and quadruple (****), respectively. All experiments were carried out in eight biological replicates.

**Figure 6 ijms-24-14302-f006:**
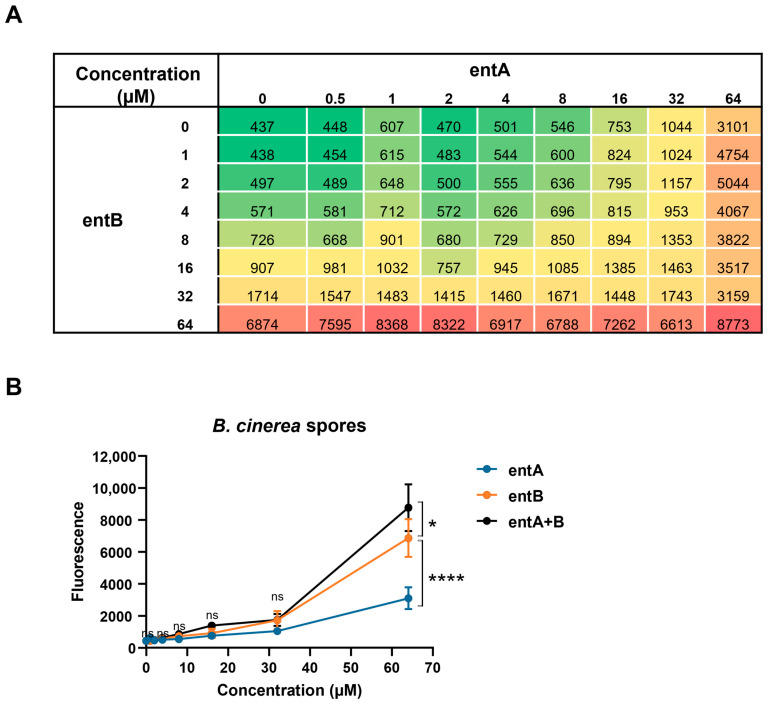
Checkerboard assay between entolysin A and entolysin B. (**A**) Checkerboard analysis of entolysin A (entA) and entolysin B (entB). The number in each cell is the mean fluorescence value of triplicates; green color indicates low activity against *B. cinerea* R16 spores, red color indicates high antifungal activity against *B. cinerea* R16 spores. (**B**) Comparison between entA, entB and entA + B (1:1 ratio) against *B. cinerea* R16 spores. The fluorescence, shown in the figure, was measured after 2 h of treatment. The vertical bars indicate the standard error (n = 3). The fluorescence values by entolysin A, entolysin B and entolysin A + B treatments were compared by two-way ANOVA analysis. Significant differences at *p* > 0.05, *p* ≤ 0.05 and *p* ≤ 0.0001 were marked as (ns), single (*) and quadruple (****), respectively.

**Table 1 ijms-24-14302-t001:** Sequence, configurational analysis and assignment of entolysin from *P. entomophila* COR5. Since entolysin A and B originate from the same NRPS system and feature the same amino acid sequence, both homologues are represented.

P. entomophila COR5	AA ^1^	AA ^2^	AA ^3^	AA ^4^	AA ^5^	AA ^6^	AA ^7^	AA ^8^	AA ^9^	AA ^10^	AA ^11^	AA ^12^	AA ^13^	AA ^14^
	Bioinformatic analysis workflow													
A-domain	Leu	Glu	Gln	Val	Leu	Gln	Val	Leu	Gln	Ser	Val	Leu	Ser	Ile
C-domain	C1	E/C	E/C	E/C	E/C	E/C	E/C	E/C	E/C	E/C	E/C	^L^C_L_	^L^C_L_	E/C
Prediction	D	D	D	D	D	D	D	D	D	D	L	L	D	L
	Chemical analysis workflow													
NMR analysis	Leu	Glu	Gln	Val	Leu	Gln	Val	Leu	Gln	Ser	Val	Leu	Ser	Ile
Marfey’s analysis	D/L	D	D	D/L	D/L	D	D/L	D/L	D	D/L	D/L	D/L	D/L	L
	Synthesized (14:5) sequences *													
	Leu	Glu	Gln	Val	Leu	Gln	Val	Leu	Gln	Ser	Val	Leu	Ser	Ile
l-Ser13 (**1**)	L	D	D	D	D	D	D	D	D	D	L	L	L	L
l-Ser-10 (**2**)	L	D	D	D	D	D	D	D	D	L	L	L	D	L
d-Ser10-d-Ser13 (**3**)	L	D	D	D	D	D	D	D	D	D	L	L	D	L

* The nomenclature of the synthetic compounds is based on the (l:m) notation of the Entolysin group (14:5), where the length ‘l’ represents the total number of amino acids, while ‘m’ indicates those involved in the macrocycle. Underlined residues indicate those that are part of the macrocycle. AA ^1–14^ indicates the position of the amino acids in the sequence.

## Data Availability

All study data are included in the article and SI Appendix. NMR data for (C–H)α spectral fingerprint matching can be found at https://www.rhizoclip.be (accessed on 10 September 2023).

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
