# Peer review of "Stereomeric Lipopeptides from a Single Non-Ribosomal Peptide Synthetase as an Additional Source of Structural and Functional Diversification in Pseudomonas Lipopeptide Biosynthesis"

_ijms, 2023, doi:10.3390/ijms241814302_

Round 1
Reviewer 1 Report
The manuscript by Muangkaew et al. describes the discovery of stereo-isoforms of lipopeptides synthesized by a single NRPS. The authors used a combination of bioinformatic, synthetic and NMR studies to conclude that entolysin A and B are identical in sequence and only differ in D/L stereoisomers of one amino acid. They also showed that entolysins are produced in the ratio of 3:2 by several strains of P. entomophila and bioinformatically alluded to the conclusion that they are produced by a single NRPS with E/C-type domain with altering epimerase activity.
Several questions are below.
It is not very clear how the ratio 3:2 of the stereo-homologs was actually determined. How consistent was it between strains? If it is consistent, how could such ratio be controlled and maintained during biosynthesis? Could it have a biological role? Why was 1:1 ratio used for synergy studies?
In the section 2.4, the authors say that the deletion of etlC does not change the impact in cucumber-Pythium ultimum pathosystem. How does this mutation impact the biosynthesis and the product? It is not clear what this observation contributes to the section here.
The figure 1 needs statistical analyses. Even if this difference is statistically significant, is it biologically significant? How were the concentrations selected for the membrane permeability assays? What are the natural yields of the compounds? Most of figure 2 seems to repeat Fig 1A. Fig2 cannot be called “Synergistic activity” as such was not observed, it needs to be renamed to something like “Checkerboard assay”
Related to L. 467-468, are there any precedents of stereo-homologs produced by a single NRPS in any other organisms or for any other compounds? If not, then the statement could be stronger.
The format of the methods section is not consistent. For all the media (M9, CM), either brand, or composition, or reference need to be provided. The NMR is described in two different sections, may want to consolidate. The section 4.7 of the methods needs to describe the data analyses.
Although the manuscript is overall well written, I find the style somewhat cumbersome and in places difficult to read. I would recommend streamlining it. There are several error messages (reference source was not found) in the text. All the supplementary information references need to be specific, i.e. indicate a specific figure or table. The legends in the supplementary file need to be detailed and explanatory.
Other comments:
1. Abstract L. 17 Spelling out E/C would help here.
2. L.82 How is this alternative aa selected? Is it known?
3. L.131 The statement “highly homologous” requires supporting data and explanations.
4. L.165 The sentence needs editing
5. It would help if the legend for table 1 explains that the sequence represents both entolysin A and B.
6. L.270 It is difficult to follow and navigate in Table 1, as entolysins A and B are not indicated in the table.
7. L.277 Which figure is it?
8. L.302, 304. The underlined need to be explained.
9. L.328 Which figure is it?
10. L. 343 This needs to be either described in details or shown in a figure.
11. L. 349-351 It is not clear how the different environments would be supportive of any relevance of the observed differences.
12. L. 359 The sentence is difficult to follow
13. L.376-386 The paragraph repeats the figure legend
14. All the Latin names of species need to be italicized.
15. L.543 What was the sample prep?
16. L.619 The sentence needs editing
17. The title of section 4.6 in the Methods shall include both cells and spores.
18. L.645 Mycelium does not fit here, as the section is about spores.
Reviewer 2 Report
This is a very impressive paper that includes multiple studies, at least two of which could possibly be stand-alone papers. The authors fully explain the importance of CLiPs and cleverly use NMR together with bioinformatics to better understand their stereochemistry as well as how CLiP diversity is generated. The authors even report data suggesting potential uses for CLiPs as fungicides.
Given the sheer volume and significance of the results reported, it is perhaps not surprising that some details get lost in the shuffle. Perhaps the techniques utilized in the reported studies and the abbreviations used in this report will be "common knowledge" to someone whose expertise is specific to the field of non-ribosomal synthesis of lipopeptides, but scientists whose expertise lies elsewhere (soil chemists, NMR spectroscopists, bioinformatics specialists) may also be interested in this paper. While a non-specialist will be able to figure out most of what they need to know based on what you wrote, it might help your paper reach a broader audience if you are a wee bit more careful to clearly define domain specific terms ("E/C-type domain", "NMR fingerprinting") as soon as you use those terms, and be a little bit more detailed in your Methods section.
More specifically ...
The abstract contains a lot of detail concerning the results of NMR and bio-informatics analysis but does not even mention the results of your assays on fungal cells. I also am concerned that interested readers might not know what an "E/C-type domain" is or what you mean by "idle epimerization": I was not aware of these terms prior to reading this manuscript, although I could figure out what those terms meant from the abstract itself. I tried to think up some more constructive suggestions for editing your abstract, but I am at a loss to figure out what to cut to make room for brief definitions of key terms and for a sentence describing your results with the fungal cell leakage assays. If you cannot figure out how to modify your abstract, that is fine, but I would recommend you give some thought as to how to revise your abstract to attract a broader readership to this important paper.
I am on the fence about recommending you add "E/C-type domain" as a keyword. Have you considered doing so?
I am a little confused by Figure 3. The consensus sequence in the top half seems to be PLQ(E/A)GILYHHLSA*EQGDPYV while in the bottom half it seems to be PLQEG(L/I)LY*HHLTAD(X)GDPYQ. Shouldn't there be a gap in the top half of the figure between the Y and HH as well as a gap between the A and the D in the bottom half of the figure so the sequences align?
I don't think HRMS is as common of an abbreviation as, say, NMR. I recommend revising line 529 to read "High-resolution mass spectra (HRMS) for [...]" so readers know the meaning of the abbreviation "HRMS" (although the meaning is, I guess, obvious enough from context).
The heading "NMR" on line 542 should be replaced with "NMR and HRMS (TOF)" -- merging the sections on NMR and on HRMS (TOF) beginning on line 528. Alternatively, move the material on HRMS (TOF) in the section on NMR to the section on HRMS (TOF), perhaps also folding your general comments on NMR into section 4.5, thus moving all of the material under the NMR heading on line 542 to other sections.
In section 4.5, you should briefly describe how you obtained full assignment of the peptides: did you use any software for resonance assignment, or was it done by hand? You mention recording ROESY data: do you have a 3D structure for any of the entolysins? You may wish to consider depositing your resonance assignments in the BMRB or other suitable data repository, if you have not already done so; if you have, please include the databank and ID/accession # for your assignments.
